# Finite Element Study of PEEK Materials Applied in Post-Retained Restorations

**DOI:** 10.3390/polym14163422

**Published:** 2022-08-22

**Authors:** Hao Yu, Zhihong Feng, Ling Wang, Senay Mihcin, Jianfeng Kang, Shizhu Bai, Yimin Zhao

**Affiliations:** 1State Key Laboratory of Military Stomatology & National Clinical Research Center for Oral Diseases & Shaanxi Key Laboratory of Stomatology, Digital Dentistry Center, School of Stomatology, The Fourth Military Medical University, Xi’an 710032, China; 2State Key Laboratory of Military Stomatology & National Clinical Research Center for Oral Diseases & Shaanxi Key Laboratory of Stomatology, Department of Prosthodontics, School of Stomatology, The Fourth Military Medical University, Xi’an 710032, China; 3State Key Laboratory for Manufacturing Systems Engineering, Xi’an Jiaotong University, Xi’an 710032, China; 4Department of Mechanical Engineering, Izmir Institute of Technology, Izmir 35433, Turkey; 5Jihua Laboratory, Foshan 528000, China

**Keywords:** polyetheretherketone, finite element analysis, post-retained restoration, biomechanical behaviors

## Abstract

Background: This study aimed to investigate the biomechanical behaviors of polyether ether ketone (PEEK) and traditional materials (titanium and fiber) when used to restore tooth defects in the form of prefabricated post or customized post via computational modelling. Methods: First, the prototype of natural tooth, and the prototypes of prefabricated post and customized post were established, respectively, whilst the residual root was restored with dentin ferrule using reverse engineering methods. Then, the stress and strain of CFR-PEEK (PEEK reinforced by 30% carbon fiber) and pure PEEK (PEEK without any reprocessing) post were compared with those made in traditional materials using the three-dimensional finite element method. Results: From the stress point of view, compared with metal and fiber posts, CFR-PEEK and pure PEEK prefabricated post both demonstrated reduced post-core interface stress, post stress, post-root cement stress and root cement stress; moreover, CFR-PEEK and pure PEEK customized post demonstrated reduced post stress, post-root cement stress and root cement stress, while the strain of CFR-PEEK post was the closest to that of dentin. Conclusions: Compared with the traditional posts, both the CFR-PEEK and pure PEEK posts could reduce the risk of debonding and vertical root fracture, whether they were used as prefabricated posts or customized posts, but the biomechanical behavior of carbon fiber-reinforced CFR-PEEK restorations was the closest to dentin, no matter if they were used as prefabricated post or customized post. Therefore, the CFR-PEEK post could be more suitable to restore massive tooth defects. Pure PEEK needs filler reinforcement to be used for post-retained restoration.

## 1. Introduction

In the clinic, the treatment of a massive tooth defect generally requires root canal treatment (RCT) and post-retained restoration, and extraction in serious cases. If the elastic modulus of the post-core material is different from that of human dentin, it cannot match the physiological mobility of the tooth [1]. Therefore, the placement of a post may lead to stress concentrations in key areas and cause complications [2,3]. The elastic modulus of a traditional metal post, ceramic post and fiber-reinforced composite post (fiber post) is greater than that of dentin, hence, complications such as debonding and root fracture usually happen [1,4,5,6,7]. The ideal post-core material requires an elastic modulus similar to that of dentin to facilitate the uniform transmission of occlusal stress in the restoration and the tooth tissue [8].

However, there is no consensus on the ideal application form of post and core materials. Some researchers have contended that the post should be prefabricated and separated from the core, while others believe that the post and core should be integrated as a whole and customized according to the root canal [9,10]. In that way, the bonding interface and stress concentration between the post and the core could be avoided [10,11]. However, if the elastic modulus of the restoration material is as high as in metal and ceramic materials, then the customized post will cause catastrophic failure such as vertical root fracture [12,13]. Therefore, although one-piece post and core seem to be an ideal form of post-retained restoration, its material should provide an optimum solution for this combination. 

Polyetheretherketone (PEEK) is a compound composed of ether and ketone group and has excellent comprehensive performance [14,15]. The elastic modulus of pure PEEK material is relatively smaller (3–4 GPa), and its tensile strength is 80 MPa, while the elastic modulus of carbon fiber reinforced-polyetheretherketone (CFR-PEEK) can reach to 18 Gpa, and its tensile strength is 120 MPa, which is very close to human dentin (elastic modulus: 18.6 Gpa, tensile strength: 104 MPa) [14,15,16]. Nowadays, PEEK materials are not only casted by the traditional lost-wax method, but also by CAD/CAM milling, injection molding and 3D printing [14,15,16,17]. In the field of dentistry, many researchers regard PEEK as an important new prosthetic material, which has already been used in fixed prosthesis, removable prosthesis, and implant prosthesis [1]. However, there is no report about PEEK material used for post-retained restoration so far. 

The post-retained restoration is a complex mechanical system, and the stress distribution in the structure is multi-axis and non-uniform [16]. In this study, we aimed to investigate the biomechanical behavior of PEEK and traditional materials (titanium and fiber) when they are used to restore tooth defects in the form of a prefabricated post or customized post, using the finite element modelling technique, in order to provide insight based on evidence for their clinical application. Our hypothesis is that PEEK or its composites could serve as a more suitable material for post-retained restoration when compared to currently used traditional materials. 

## 2. Materials and Methods

### 2.1. Generation of the Geometric Models and Study Design

The ethical approval for this study was obtained from the ethics committee of the Third Affiliated Hospital of Air Force Medical University (Permit Number: AF-SOP-008-3.0) (Xi’an, China). Mimics software (Mimics research 20.0; Materialise, Leuven, Belgium) was used to extract the Cone Beam Computed Tomography (CBCT) data from volunteers, and one left maxillary second premolar which met the standard size of Chinese normal maxillary second premolar was selected [18]. The models of tooth, enamel and pulp were extracted and saved as STL files. Since there were many surface irregularities consisting of concaves or convexes, the preliminary extracted models were smoothed by Geomagic software (Geomagic Studio 2014; Geomagic Inc., Morrisville, NC, USA). Then, these models were converted to IGS files and imported into the finite element software (ANSYS Workbench 17.0; ANSYS Inc., Canonsburg, PA, USA).

It takes a great amount of time for a computer to build and calculate a model with a real anatomical size and occlusion mode. Therefore, it is often necessary to simplify some complex model structures [19]. The shell surface was defined by extracting 0.2 mm of shell from the root surface below the cemento-enamel junction (CEJ) to simulate the periodontal ligament model. The cement layer thickness was set as 0.1 mm [20]. The alveolar bone was simplified as a cube around the root with a length of 20 mm. The external part was cortical bone with a thickness of 2 mm, and its upper face was 2 mm below the CEJ [21]. The rest of the internal part was cancellous bone. The fiber post model was established based on the references [21]. Finally, the models of enamel, dentin, pulp, crown, crown cement, resin core, prefabricated post, customized post, residual root with 2-mm ferrule, root cement, periodontal ligament, cortical bone, and cancellous bone were obtained by Boolean operation (Figure 1A). Three finite element (FE) Prototypes were created: Prototype 1 with natural tooth, Prototype 2 with a prefabricated post to restore the root with 2-mm ferrule, and Prototype 3 with a custom-made post and core (Figure 1B). 

The mechanical properties of each model were assigned according to the literature data (Table 1). 

It is assumed that all materials were isotropic, homogeneous and continuous [21,25]. It has been reported that PEEK materials could obtain enough bond strength with resin by pretreatment [26,27]. Therefore, in order to avoid the influence of different cements on the experimental results, the same resin cement was utilized in each group. In terms of post material selection, a titanium post and a fiber post were both utilized as traditional materials. CFR-PEEK is actually PEEK material reinforced by 30% carbon fiber, while pure PEEK is pure PEEK without any reprocessing [14]. 

### 2.2. Finite Element Analysis (FEA)

All contact surfaces are defined as perfectly bonded. Through the mesh sensitivity analysis, the element size of three prototypes was set to 0.5 mm. The orthogonal quality was all above 0.85, which indicated that the mesh quality was favorable (Figure 2A–C). The bottom of the alveolar bone was set as a fixed support in clinical practice. The average chewing force of maxillary second premolars is 105 N [28], and the functional load was applied to the buccal incline of the palatal cusp at 45° to the long axis of the tooth (Figure 3). In this way, we could evaluate the biomechanical behavior of post-core materials under normal occlusion load more practically.

According to the material property, the Maximum Principal Stress (MPS) was used to evaluate the stress of each prototype [29]. The root stress in Prototype 1 was used as a reference to evaluate the effect of different restorations on the root fracture resistance. As the bond strength of the bonding interface and the tensile strength of all the materials were not available, the greater stress value for the same model meant a higher failure risk [20]. It was shown that the stress of the bonding interface was an important factor affecting the bonding failure and root fracture [2,4,13]. Therefore, the stress of the bonding interface was used to evaluate the risk of post debonding [2,4,13]. In this experiment, the strain value of a prefabricated post and a customized post were analyzed, respectively, and their difference values with roots were compared. The smaller absolute value of these differences indicated that restoration had more similar biomechanical behavior with root, which was more conducive to the stability of the whole restoration system.

## 3. Results

The peak values of stress and strain were recorded, and the stress distribution was observed. In Prototype 1, the root stress was 23.25 MPa, and concentrated in the 1/3 area of the neck of the root, decreasing gradually from outside to inside (Figure 4A). In Prototypes 2 and 3, the stress distribution of root was similar to the Prototype 1 (Figure 4B,C). In Prototype 2, the post-core interface stress, post stress, post-root cement interface stress and root cement stress all decreased with the decrease in the posts’ elastic modulus (Figure 5). The stress of CFR-PEEK and pure PEEK prefabricated posts to the resin core was smaller than that of traditional posts, while the core stress of the pure PEEK post was larger than that of the CFR-PEEK post (7.4%) (Figure 5). In Prototype 3, the stress of the customized post, root cement and post-root cement interface all decreased in line with the decrease in restorations’ elastic modulus (Figure 6). Additionally, in Prototypes 2 and 3, the root stresses increased slightly (less than 10%) with the decrease in the elastic modulus of restorations (Figure 5 and Figure 6).

From titanium to pure PEEK, with the decrease in the elastic modulus, the stress concentration of the post, post-root cement interface and root cement all became less and less in Prototypes 2 and 3 (Figure 7, Figure 8 and Figure 9). Moreover, in Prototype 2, the titanium post, fiber post and pure PEEK post had a certain stress concentration in the core which contacted with the post head, but the stress concentration of the CFR-PEEK post to the core was unobvious (Figure 10). With the decrease in the post elastic modulus, the stress concentration of the post-core interface became increasingly less in Prototype 2 (Figure 11). However, the root stress concentration in these three prototypes was similar and all concentrated in the 1/3 area of root neck, decreasing gradually from outside to inside (Figure 4).

From the perspective of strain in Prototypes 2 and 3 (Table 2), the strain of titanium restoration, fiber post restoration and CFR-PEEK restoration were all smaller than the corresponding root strain, but the absolute value of strain difference of the CFR-PEEK post was the smallest. The strain of the pure PEEK post itself was the largest, even larger than that of the root, and the absolute value of the strain difference was also higher than that of the CFR-PEEK post both in Prototype 2 and Prototype 3.

## 4. Discussion

Debonding is reported as the common failure cause of post-retained restoration, especially for the prefabricated posts [12,13,29]. If the post and core were made separately, it would inevitably increase the bonding interface and debonding risk [12,13,29]. In Prototype 2, the core stress and post-core interface stress of the titanium post and fiber post were higher than that of the PEEK posts, and noticeable stress concentration areas were detected. Higher stress in the core means a higher risk of fracture and crown adhesion failure [30]. Therefore, the post-core complex was vulnerable when the titanium and fiber posts were loaded, which corresponds to the clinical situation that these two kinds of posts are more prone to separation from the core easily [8]. The CFR-PEEK and pure PEEK prefabricated posts could reduce the core stress and stress concentration. However, compared with the CFR-PEEK post, the pure PEEK post would slightly increase the stress of the resin core by 7.4% and increase the risk of crown debonding compared with CFR-PEEK post.

According to the literature [8,20,31], the interface between the post and cement was more likely to fail than that between cement and dentin, which was not only related to the bonding strength of material itself, but also related to the mismatch of the elastic modulus between the post and root. In Prototypes 2 and 3, the post-root cement interface stress and root cement stress of two PEEK materials were significantly smaller than those of traditional materials. Moreover, the stress concentration points were difficult to detect. Smaller and less stress concentration regions in the bonding interface and root cement meant a lower risk of debonding [20,31]. Therefore, CFR-PEEK and pure PEEK material might be more beneficial to reduce the post debonding risk than the traditional materials.

Most studies have shown that the main role of the post and core is to connect the crown and root as a whole, and the most important factor affecting root fracture is the amount of residual tooth tissue after RCT [5,6,32,33]. Prototypes 2 and 3 simulated the tooth defect with a 2-mm ferrule, which was ideal for clinical practice [8], so the root stress of the four materials in these two prototypes was approximate and all less than the natural root stress (23.25 MPa) in Prototype 1. Moreover, the stress distribution of the root was almost the same in these three prototypes (Figure 4). Thus, CFR-PEEK and pure PEEK may not affect the fracture resistance of residual root whether they are used as a prefabricated post or a customized post.

Considering that debonding was the initial cause of root fracture, the area in which debonding first occurred would affect the type of root fracture [13,29]. When the stress concentration led to the bonding interface failure, the wedge-like post would contact the inner wall of the root canal directly and conduct downward and outward force to the residue root, causing vertical root fracture [12]. Moreover, this vertical root fracture was difficult to restore twice and the root could only be extracted. In Prototypes 2 and 3, the bonding interface stress and root cement stress of titanium restorations mainly concentrated on the tip and in the middle (1/3 area of the post), which corresponded to the occurrence of “wedge effect” (Figure 8 and Figure 9). CFR-PEEK and pure PEEK had a smaller elastic modulus and, thus, not only were their post-root cement stress and root cement stress significantly smaller, but the stress distribution was also more uniform. Therefore, the CFR-PEEK and pure PEEK restorations would help reduce the risk of vertical root fracture. In this experiment, the debonding risk of CFR-PEEK and pure PEEK restoration was relatively lower, so even if the local area was debonded, as the elastic modulus of these two materials was not higher than that of dentin, it would probably not cause catastrophic failure.

To evaluate which one of these two PEEK materials had better biomechanical behavior, the strain values should be inspected. The absolute value of the difference between the strain of CFR-PEEK restorations and the strain of the root was the smallest in both Prototype 2 and Prototype 3, which indicates that the biomechanical behavior of CFR-PEEK restorations was more favorable. When the tooth was loaded, CFR-PEEK prosthesis could obtain consistent physiological mobility with the root. The ideal post material should not only be close to the dentin in terms of the elastic modulus, but also be able to resist a certain level of deformation [8]. However, the strain of pure PEEK restoration was larger than that of its root, which showed that this material was relatively softer and had lower fracture resistance. 

In the clinic, dentin ferrule, root canal preparation, post material and shape, core material and retention form, cement material and thickness, crown material, etc., can affect the success of post-retained restoration. This study could not set the same temperature and humidity as the oral environment, nor could it accurately simulate the actual chewing force and assess whether the restoration was successful. However, by constraining most variables mentioned, this study could evaluate the failure risk of different post and core materials, which was more controllable and intuitive than clinical trials. This experiment proves that the PEEK post was theoretically helpful in reducing the risk of debonding and vertical root fracture when compared with traditional materials. The biomechanical behavior of CFR-PEEK material was closest to that of the root, which was beneficial for the long-term stability of the prosthesis and the root. Therefore, the hypothesis tested in this experiment turned out to be valid.

Some studies have shown that the bond strength of PEEK and inorganic fiber is higher than that of epoxy resin matrix and inorganic fiber, and the mechanical strength and biocompatibility of the compound are better [34,35]. Moreover, PEEK materials can be manufactured by casting, CAD/CAM, injection molding and 3D printing, which can nicely meet the requirement of personalized oral restorations [14,15,16,17]. Therefore, a PEEK compound could be more suitable for fabricating a root post. Due to the above limitations of this study, long-term clinical trials are still needed, but the results of this study can provide a theoretical basis.

## 5. Conclusions

Giving the limitations of this finite element study, the following conclusions can be drawn:Compared with traditional posts, the CFR-PEEK and pure PEEK posts could both reduce the risk of debonding and vertical root fracture. However, the biomechanical behavior of the carbon fiber-reinforced CFR-PEEK restorations was the closest to dentin, no matter if it was used as a prefabricated post or customized post. Therefore, the CFR-PEEK post could be more suitable to restore massive tooth defects;Compared with CFR-PEEK, the pure PEEK post would increase the stress of resin core when used as a prefabricated post and is probably unable to withstand bite force. Pure PEEK needs filler reinforcement to be used for post-retained restoration.

## Figures and Tables

**Figure 1 polymers-14-03422-f001:**
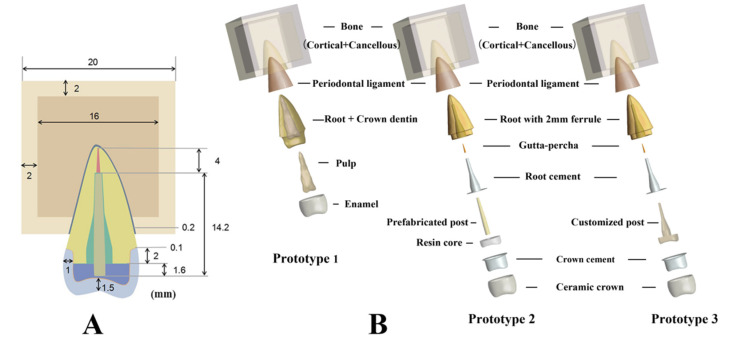
Dimension and establishment of three finite element Prototypes. (**A**) Dimension of Prototypes. (**B**) Structure of three Prototypes.

**Figure 2 polymers-14-03422-f002:**
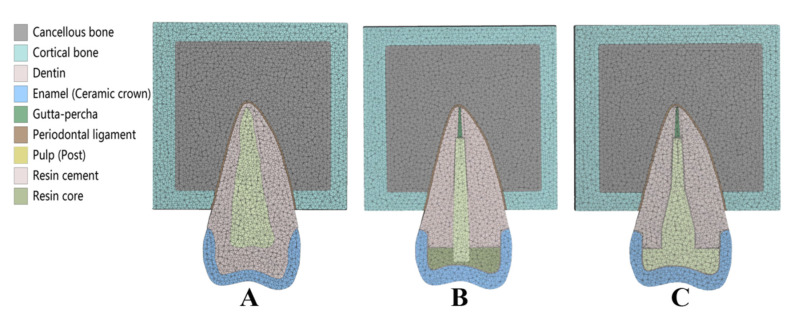
Mesh of Prototypes. (**A**) Mesh of Prototype 1. (**B**) Mesh of Prototype 2. (**C**) Mesh of Prototype 3.

**Figure 3 polymers-14-03422-f003:**
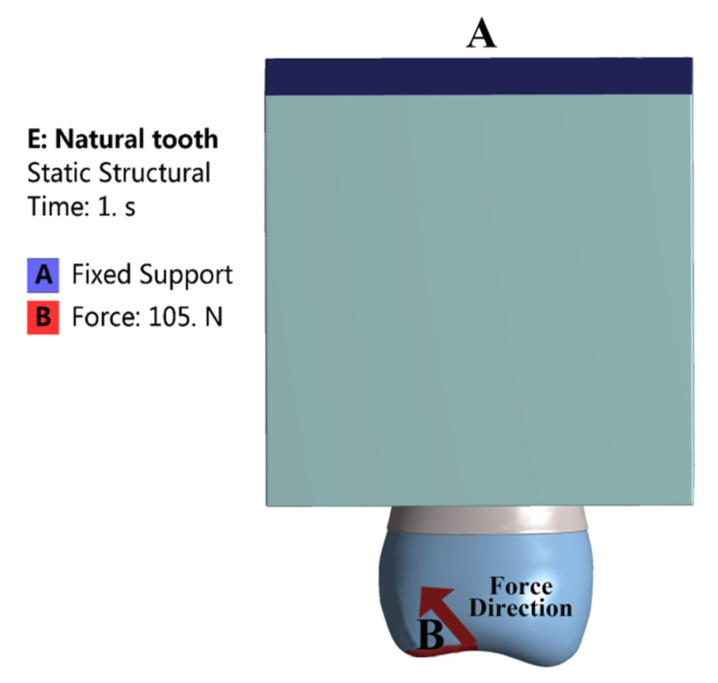
Boundary conditions of Prototypes.

**Figure 4 polymers-14-03422-f004:**
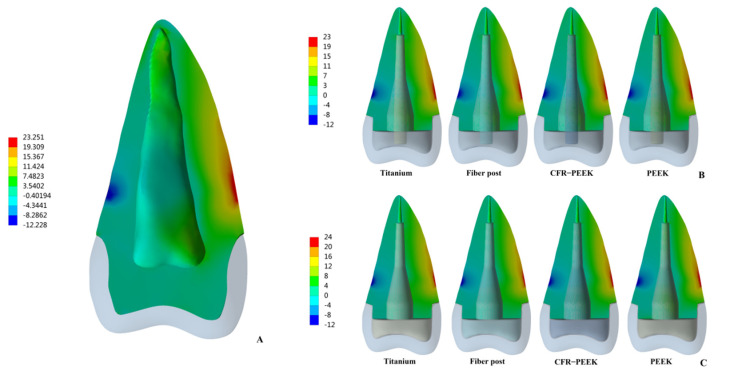
Distributions of MPS at roots. (**A**) Prototype 1. (**B**) Prototype 2. (**C**) Prototype 3.

**Figure 5 polymers-14-03422-f005:**
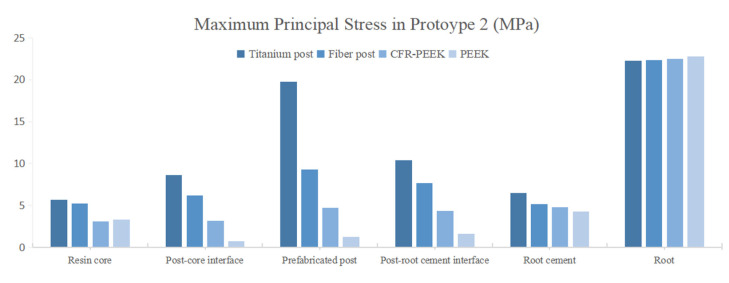
Peak value of MPS at post in Prototype 2.

**Figure 6 polymers-14-03422-f006:**
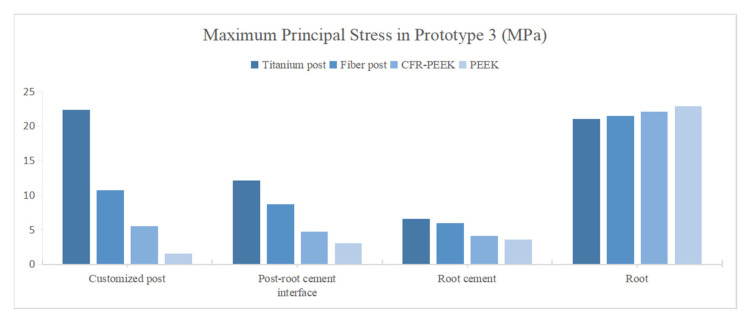
Peak value of MPS at post in Prototype 3.

**Figure 7 polymers-14-03422-f007:**
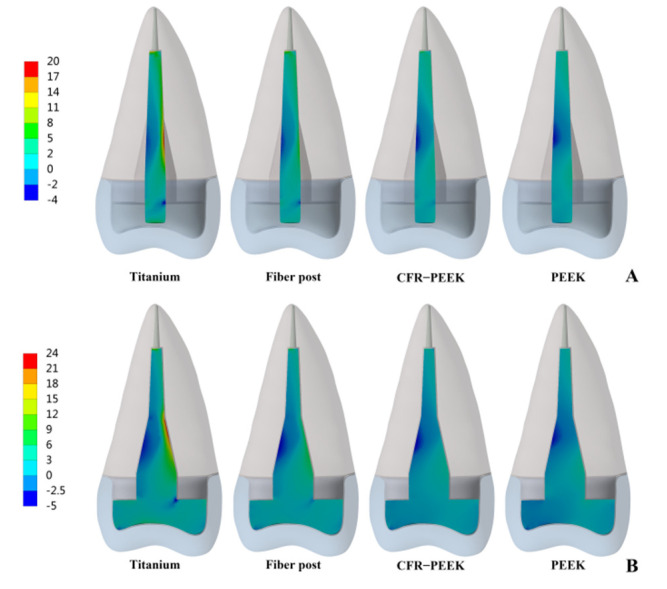
Distributions of MPS at post. (**A**) Prototype 2. (**B**) Prototype 3.

**Figure 8 polymers-14-03422-f008:**
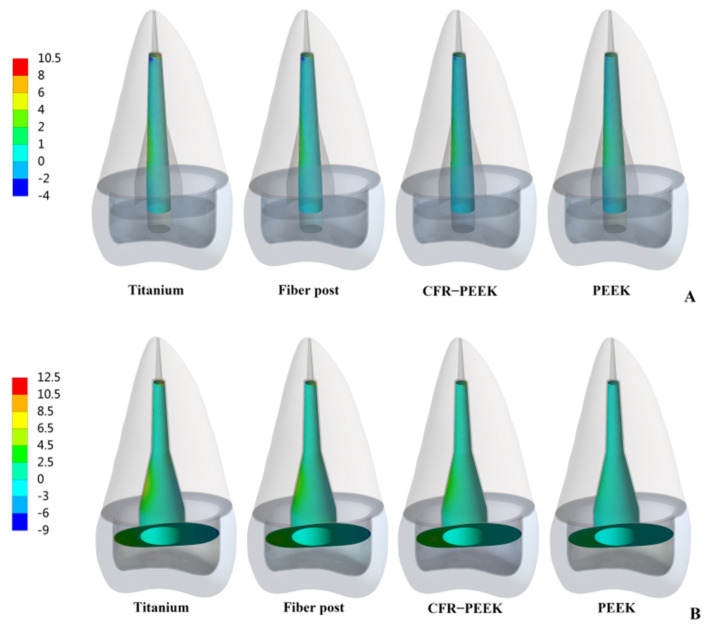
Distributions of MPS at post-root cement interface. (**A**) Prototype 2. (**B**) Prototype 3.

**Figure 9 polymers-14-03422-f009:**
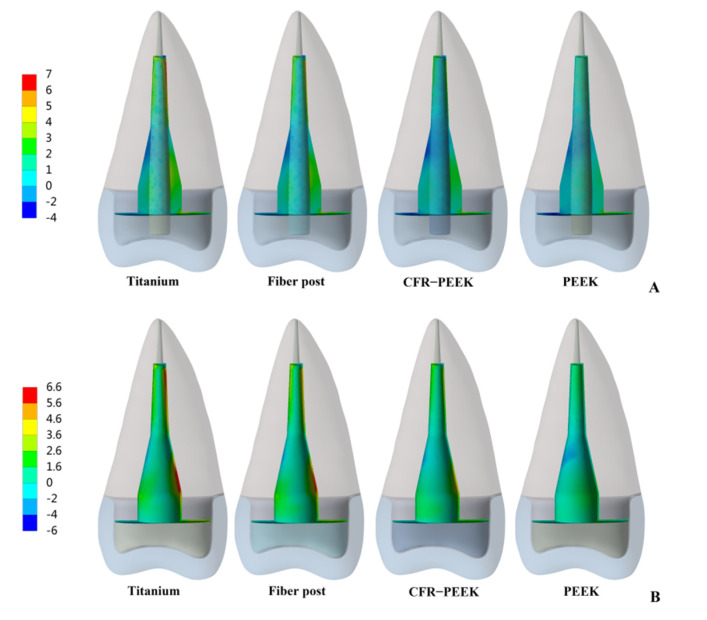
Distributions of MPS at root cement. (**A**) Prototype 2. (**B**) Prototype 3.

**Figure 10 polymers-14-03422-f010:**
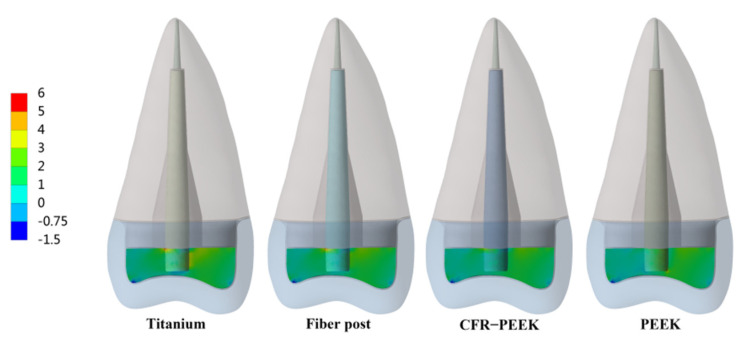
Distributions of MPS at resin core in prototype 2.

**Figure 11 polymers-14-03422-f011:**
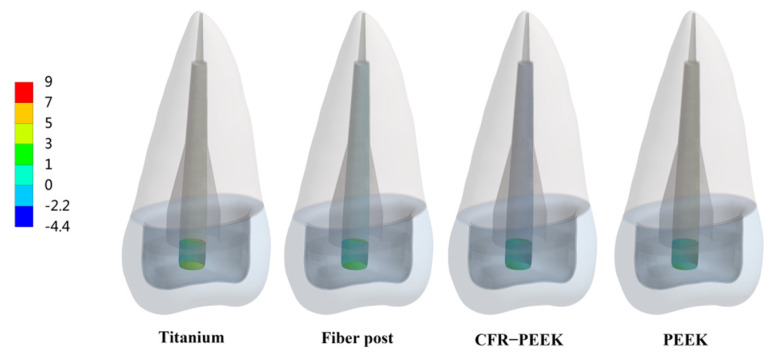
Distributions of MPS at post-core interface in prototype 2.

**Table 1 polymers-14-03422-t001:** Mechanical properties of each material.

Material	Young’s Modulus (Gpa)	Poisson Ratio	Reference No.
Enamel	84.1	0.33	[22]
Dentin	16.8	0.31	[4]
Pulp	0.02	0.45	[4]
Periodontal ligament	0.069	0.45	[4]
Cortical bone	13.7	0.3	[4]
Cancellous bone	1.37	0.3	[4]
Ceramic crown	62.0	0.3	[4]
Resin cement	5.0	0.3	[4]
Resin core	20.0	0.3	[4]
Gutta-percha	0.69	0.45	[4]
Titanium post	120	0.3	[23]
Fiberglass post	53.8	0.3	[4]
CFR-PEEK	18.0	0.39	[24]
Pure PEEK	4.1	0.4	[25]

**Table 2 polymers-14-03422-t002:** Strain of root and post.

Group	Material	Root Strain	Post Strain	Absolute Value of Difference
Prototype 2	Titanium post	4.10 × 10^−4^	1.63 × 10^−4^	2.47 × 10^−4^
	Fiber post	3.62 × 10^−4^	1.65 × 10^−4^	1.97 × 10^−4^
	CFR-PEEK	3.77 × 10^−4^	2.95 × 10^−4^	8.20 × 10^−5^
	Pure PEEK	4.12 × 10^−4^	5.59 × 10^−4^	1.47 × 10^−4^
Prototype 3	Titanium post	4.06 × 10^−4^	1.90 × 10^−4^	2.16 × 10^−4^
	Fiber post	3.71 × 10^−4^	1.71 × 10^−4^	2.00 × 10^−4^
	CFR-PEEK	3.52 × 10^−4^	2.78 × 10^−4^	7.40 × 10^−5^
	Pure PEEK	4.39 × 10^−4^	5.29 × 10^−4^	9.00 × 10^−5^

## Data Availability

The data presented in this study are available on request from the corresponding author.

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
