# Peer review of "Finite Element Study of PEEK Materials Applied in Post-Retained Restorations"

_polymers, 2022, doi:10.3390/polym14163422_

Round 1

Reviewer 1 Report

This is an interesting article looking at stress distribution in dental restorations of different materials and two different designs.  The lower modulus PEEK and CFR PEEK materials are better at transferring stress away from the implant/cement interface and into the root which may reduce the potential for debonding of the implant.  This is solely an FE investigation with no experimental validation.

General comments

To make the article more accessible to a more general polymers audience, it would help for some of the specific dentistry terms to be clarified.  For example, can more description of the materials for the fibre posts be given, can abbreviations such as CBCT be written in full in their first use.  The article would benefit from thorough proof reading.  There seem to be some issues with the figure numbers, please check.

Title

Please consider altering the title to highlight that this is an FE/in silico study

Introduction

Lines 61-63: Injection moulding is also being used to fabricate PEEK femoral components

Materials and Methods

Lines 78-82: details relating to the 5 subjects scanned do not seem relevant to the study as only 1 tooth geometry was chosen from the scanned subjects.  This information should be removed.

Please describe the boundary conditions used in the model, were any other boundary conditions used in addition to the fixed support at the base of the bone?  What did the magnitude and direction of the applied force represent?

Table 1: can more the materials for ‘ceramic crown’ and ‘fiber post’ be more specific

Line 104: this information is duplicated, please remove this line

Figure 1: please label the root in prototype 1.  This figure is helpful, could the text and the force direction be made larger?

Line 154: lists figure 10 but this is not in the manuscript, please check the figure numbers

Line 156: should the figure number on this line read figure 2?

Line 164: please check this figure number

Table 2: could the root strain for prototype 1 be added to table 2?

Discussion

Given the assumptions made in this simulation, there are a number of limitations, an additional paragraph stating these and further experimental validation required which would help to put the work into context. 

Aside from the mechanical properties of the root, please describe other properties/features which would influence bonding for example the surface topography and also consider that when changing materials, a change in implant geometry may also be necessary, please describe this. 

The study measured stress & strain whilst these may contribute to debonding to the implants, there are many other factors which influence fixation including the heat dissipation of the material, the surface finish, etc.  Please be careful in describing the stress/strain distribution as debonding.    

It would also help to put these results into context by looking to other research of PEEK implants.  For example de Ruiter et al has carried out similar work looking at a PEEK femoral component for total knee replacement and compared to a CoCr implant, his work on stress distribution (2017) and fixation (Materials, 2020) is relevant to this study and would help to put it in context.

Line 239: please add injection moulding to the list of manufacturing techniques for PEEK

Author Response

Dear teacher:

Thank you for your comments. We appreciate it very much for your constructive comments and suggestions on our manuscript entitled ''Biomechanical study of PEEK materials applied in post-retained restorations'' (ID: polymers-1840205) .Those comments are all valuable and very helpful for revising and improving our paper,as well as the important guiding significance to our researches. I have revised the article as your suggestions.Please see the attachment

Kind regards,
Mr. Yu
Author
E-Mail: [email protected]
  Mr. Bai Correspond author E-Mail: baishizhu@foxmail. com

Reviewer 2 Report

The manuscript “Biomechanical study of PEEK materials applied in post-retained restorations” deals with an actual problem related to design of dental prosthesis. The investigates biomechanical behaviors of polyether ether ketone (PEEK) in contrast to traditional materials (titanium and fiber) being used for restoration of tooth defects; for doing so, prefabricated post or customized post were computer simulated. The stress and strain of CFR-PEEK and pure PEEK posts were compared with the help of three-dimensional finite element method. It was shown that when compared with the traditional posts, both the CFR-PEEK and neat PEEK posts could reduce the risk of debonding and vertical root fracture in the case of the prefabricated posts or customized posts. At the same time, the CFR-PEEK post is more appropriate for the restoration of massive tooth defects.

- The paper falls within the scope of the journal of Polymers.

- The manuscript is of due size.

- The state of the art is clearly described with the number of cited papers equal to 33. Most of the references are modern and relevant.

- The Materials and Methods section is clearly written. The simulation might be reproduced elsewhere.

- The results are reported with numerous graphs and MPS-distributions.

- The Discussion is given as a separate section. The computational evidences have been duly interpreted.

- The Conclusion is very brief and must be broadened.

- The level of the English language is OK.

The manuscript requires minor-to-major revision. The following issues are to be addressed.

1Page 2, line 40. “it is unable to simulate the physiological activity of tooth [1].”. Change the style, please.

2Page 3, line 80. What does the CBCT mean?

3Page 4, line 104. Please remove an excessive phrase “It is assumed that..”

4Page 5. The authors should illustrate the appearance of FEM-mesh as a separate figure. It would be a very important illustration.

5Page 5, figure 1. The selection of the load value (Force) and direction is to be substantiated.

6Page 5, line 122. “According to the material property”. What material and what property are meant?

7Page 5, line 127. It should be: “It was shown”!

8Page 9. Table 2. The numbers should be presented in a better, readable form. It is very difficult to analyze them with a huge number of zeroes.

   Page 10. Line 221. “To decide which one of these two PEEK materials had better biomechanical behavior. Strain values should be inspected”. It seems that an extra dot has been inserted.

   The conclusion is very brief. It does not look like an appropriate summary for the scientific paper. The numerical values are to be used and discussed in the Conclusion.

   It is not very clear, what is a fiber post? More detailed description is required.

Author Response

Dear teacher:

Thank you for your comments. We appreciate it very much for your constructive comments and suggestions on our manuscript entitled ''Biomechanical study of PEEK materials applied in post-retained restorations'' (ID: polymers-1840205) .Those comments are all valuable and very helpful for revising and improving our paper,as well as the important guiding significance to our researches. I have revised the article as your suggestions.Please see the attachment.

Kind regards,
Mr. Yu
Author
E-Mail: [email protected]

Mr. Bai

Correspond author

E-Mail: baishizhu@foxmail. com

Reviewer 3 Report

The work could be interesting, but due to serious errors it is not suitable for publication. I propose to correct and submit the work again.  
I give major, but I will change to reject if the convergence analysis is not correctly presented in the methodology and directional modulus of fiber posts 

1. 
In your model, fiber reinforced materials are modelled as isotropic. This disqualifies the research because the concept of their application is based on highly directional properties that determine load transfer. 

2. 
The load of 105 N doesn`t allow for strength analysis. Lateral occlusal load can reach 600-800N 

3. Line “element size of three prototypes was set to 0.5mm.” 
this size is unacceptably huge for these small elements.  
It assumes initial coarse size with 2 elements per thickness. COARSE. 
The influence of mesh density in the criteria areas has not been investigated.  
In the case of stress between materials, it is of particular importance, as the values at the nodes at the interface are "singular" and require an additional value assessment procedure, especially the peak value is an artifact of FEM. 
Special procedures are required to obtain an exact values. The mesh density increase is needed to obtain the gradient function of values from nodes away from singular node. The value in the singular node is discarded and recovered from the gradient functions from the nodes that are not singular. For this, fine mesh is needed to obtain a physical function: Fig. 3 [Å»mudzki, J., Walke, W., & Chladek, W. (n.d.). Influence of Model Discretization Density in FEM Numerical Analysis on the Determined Stress Level in Bone Surrounding Dental Implants. Information Technologies in Biomedicine, 559–567. doi:10.1007/978-3-540-68168-7_64] 
 In the case of coarse mesh there are underestimated stress and overestimated zone. Albeit the peak stress tends to infinity with finer mesh.  
Therefore, peak values cannot be a criterion, at most if the models and meshes are identical, it shows the tendency to increase or decrease, but you have different models and different meshes. The meshes that have not been checked, convergence results shown, and no meshes are displayed on the figures. 
Please show results with mesh and with enlargement of criterial zones.  
Show elemental and nodal value comparison.  
If you see that there is no convergence, then you need to do another calculation for the mesh finer 

Additional: 

Line 19 traditional materials (titanium and fiber) 
Fiber ? I think fiber reinforced composite 

Line 33 to its excellent biomechanical behaviors. 
what does it mean? You have to be precise in what criteria something is better or worse 

Line 41 physiological activity of tooth 
physiological activity? Load transfer you want to say? 

Figure 1. 
Main and important dimensions. 

Line 104 It is assumed that 
delete

Author Response

(The authors gave the same response as above.)

Reviewer 4 Report

Major flaws are -

1. Please add the ethical clearance number.

2. five male volunteers - Please add more subjects and do the comparative statistical analysis.

3. left maxillary second premolars - Please add more teeth and do the comparative statistical analysis.

4. Add p-value results.

Author Response

(The authors gave the same response as above.)
